# UNCONDITIONAL HUMAN MOTION AND SHAPE GENERATION VIA BALANCED SCORE-BASED DIFFUSION

## ABSTRACT

Recent work has explored a range of model families for human motion generation, including Variational Autoencoders (VAEs), Generative Adversarial Networks (GANs), and diffusion-based models. Despite their differences, many methods rely on over-parameterized input features and auxiliary losses to improve empirical results. These strategies should not be strictly necessary for diffusion models to match the human motion distribution. We show that on par with state-of-the-art results in unconditional human motion generation are achievable with a score-based diffusion model using only careful feature-space normalization and analytically derived weightings for the standard L2 score-matching loss, while generating both motion and shape directly, thereby avoiding slow post hoc shape recovery from joints. We build the method step by step, with a clear theoretical motivation for each component, and provide targeted ablations demonstrating the effectiveness of each proposed addition in isolation.

## 1 INTRODUCTION

In this work, we show that a score-based diffusion model can generate unconditional motion on par with state-of-the-art methods without auxiliary regularization losses encoding motion priors, without redundant human-motion representations, and without slow post-processing for shape. Our approach combines a principled weighting of the standard score-matching L2 loss with careful normalization across feature groups in a minimal, SMPL-based motion representation. We develop the method step by step, providing theoretical motivations for each weight and normalization and validating their empirical effectiveness.

Many classes of generative models have been applied to human motion generation with great empirical success, such as Variational Auto-Encoders (VAEs) (Guo et al., 2022; Rempe et al., 2021; Petrovich et al., 2022; Kingma et al., 2013), Generative Adversarial Networks (GANs) (Raab et al., 2023; Barsoum et al., 2018; Goodfellow et al., 2020) and diffusion models (Chen et al., 2023; Tevet et al., 2023; Song et al., 2020b; Ho et al., 2020; Zhang et al., 2024a). Despite their differences, these methods often rely on similar design choices that can complicate modeling and training.

One such common characteristics is to use different but redundant representations of the human motion data (Zhang et al., 2024b; Chen et al., 2023; Guo et al., 2022; Tevet et al., 2023; Rempe et al., 2021) which we term over-parameterized input features. Specific examples of this include combining absolute position with velocity, 3D joint position with joint angles or by adding foot contact labels. Another shared property is the introduction of extra auxiliary losses in training (Zhang et al., 2024b; Rempe et al., 2021; Tevet et al., 2023; Guo et al., 2022; Chen et al., 2023), to complement the losses associated with the generative training process and encourage desirable properties.

In many cases both of these concepts improve final empirical results, but they add extra complexities in training. Over-parameterized input features are entirely empirically motivated, but are difficult to analyze and understand exactly why and how they help. Having multiple training losses necessitates fiddly and time consuming hyperparameter optimization to find a good weighting of the different losses. Multiple losses might be necessary for some generative models, such as VAEs, to compensate for the assumptions made about the data distribution. For generative models from the diffusion model family, the auxiliary losses introduce several problems. At low signal-to-noise ratios (i.e. high noise levels), losses penalizing deviations from valid motions are not effective, as the optimal predictions might not be a valid motion (Karras et al., 2022). Additionally, as auxiliary losses alter the generation

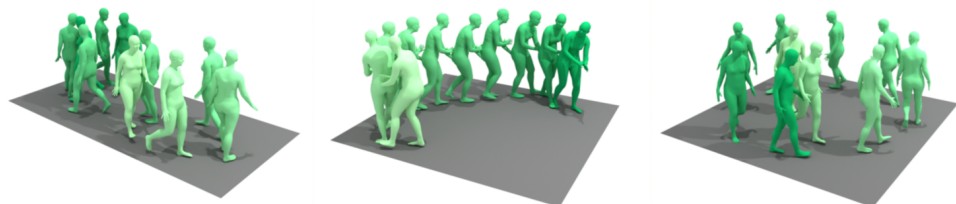

Figure 1: Unconditionally generated samples from our final model. SMPL parameters are generated directly and the mesh is extracted with the SMPL-H model. Generated with 31 NFE. Darker color indicates later frames in the sequence. See supplementary videos for more qualitative results.

vector field, we can no longer pose sampling as solving the probability-flow ODE (PF-ODE) (Song et al., 2020b). Consequently, we lose the ability to employ ODE solvers to sample from the data distribution, and instantaneous change-of-variables likelihoods (Chen et al., 2018) no longer pertain to the data.

We argue that neither over-parameterized features nor auxiliary losses should be strictly necessary to match the human motion-and-shape distribution with diffusion models. We believe many of the difficulties in matching the distribution arise from the heterogeneous feature space used to represent motion, concatenated components with different structure, statistics and dimensionality, which induces imbalances during training. To address the imbalances, we adapt and extend tools originally developed by Karras et al. (2024) for balancing training dynamics in diffusion models for images.

We deliberately focus on unconditional human motion and shape generation in an SMPL parameterization. Our goal is to model full pose trajectories, including limb axis twists that are not identifiable from 3D joint coordinates, together with global orientation, global translation, and shape. We study unconditional generation because in many conditional scenarios the conditioning information can be sparse, missing or noisy, (e.g. motion infilling with very sparse observations, a future use case outside the scope of this work) and success then relies on a strong unconditional prior.

Our main contribution is a structure-preserving feature normalization for the SMPL parameters, together with theoretically motivated weightings for the L2 score-matching loss, each evaluated in isolation. This formulation enables:

- Unconditional human motion diffusion training without empirical tuning of loss weights.
- PF-ODE compatibility for sampling and likelihoods.
- Direct shape generation (removing the need for post-hoc recovery from joints).
- Results on-par with state of the art with as few as 31 neural function evaluations (NFE).

Figure 1 illustrates typical motion and shape outputs produced under our default sampling setup.

## 2 RELATED WORKS

**Human motion diffusion models** There exists a multitude of human motion diffusion models with common applications such as text-to-motion (Chen et al., 2023; Tevet et al., 2023; Zhang et al., 2024a; Yuan et al., 2023), action-to-motion (Chen et al., 2023; Tevet et al., 2023; Zhang et al., 2024a; Yuan et al., 2023) and general purpose priors used for several downstream tasks (Zhang et al., 2024b). Prior work utilizes both stochastic ancestral samplers and deterministic DDIM-style (Song et al., 2020a) updates. Many methods parameterize the network to predict the clean sample, which makes it possible to attach auxiliary losses that encode motion priors during training or at sampling time (Tevet et al., 2023; Zhang et al., 2024a;b; Yuan et al., 2023).

**Input features in human motion generation** Human motion representations are as varied as the methods modeling them (Loper et al., 2023; Terlemez et al., 2014; Guo et al., 2022; Ionescu et al., 2013). Different datasets commonly supply their data in some format, which can roughly be divided into joint angles and joint 3D coordinates. AMASS (Mahmood et al., 2019) supplies SMPL (Loper et al., 2023) parameters containing joint angles which exist in other format as well (Terlemez et al.,

2014). Human3.6M (Ionescu et al., 2013) supplies 3D joint positions based on motion capture markers. However, these raw formats are rarely used directly. For example, the HumanML3D dataset (Guo et al., 2022) extracts their own input features, over parameterizing and combining for example joint 3D positions and joint angles. Several works also combine the SMPL parameters with 3D joint positions and foot contact labels (Zhang et al., 2024b; Rempe et al., 2021). Adding foot contact labels is a common example of over parameterization (Chen et al., 2023; Rempe et al., 2021; Zhang et al., 2024b; Guo et al., 2022; Jiang et al., 2023; Zhang et al., 2024a; Tevet et al., 2023; Raab et al., 2023).

**Auxiliary losses in human motion generation** Auxiliary losses are typically linked to the input feature representation used. When rotations are predicted, auxiliary losses can be added for 3D positions, retrieved through forward kinematics (Tevet et al., 2023; Raab et al., 2023). Losses to combat foot skating are common, restricting foot velocity depending on foot contact labels (Zhang et al., 2024b; Tevet et al., 2023). Losses for regularization of velocity in isolation are also used frequently (Tevet et al., 2023; Jiang et al., 2023; Rempe et al., 2021; Zhang et al., 2024b). In the context of diffusion-based generative models, we also consider sampling guidance to be a form of auxiliary loss. They are used for the same purposes, such as combating foot skating (Zhang et al., 2024b). Lately, methods such as PhysDiff (Yuan et al., 2023) apply sampling guidance with the help of a physics engine, trying to ensure physically correct motions.

## 3 PRELIMINARIES: EDM & EDM2

We begin with a brief overview of the work by Karras et al. (2022; 2024), which forms the foundation of our approach and is adapted here to the human motion setting. Their contributions are twofold: first, they analyzed and refined the score-based generative process, resulting in the EDM method (Karras et al., 2022); second, through a study of training dynamics, they introduced architectural improvements in the EDM2 network along with a set of standardization tools (Karras et al., 2024).

### 3.1 THE EDM SCORE-BASED GENERATIVE METHOD

The score-based generative process proposed by Karras et al. (2022) is a variance exploding continuous time diffusion process with the forward process

$$\mathbf{x}(t) = \mathbf{x}(0) + t\epsilon \tag{1}$$

where $\mathbf{x}(0)$ is a sample from the data distribution and $\epsilon \sim \mathcal{N}(\mathbf{0}, \mathbf{I})$. This leads to the corresponding PF-ODE

$$d\mathbf{x}(t) = -t \, \nabla_{\mathbf{x}(t)} \log p(\mathbf{x}(t), t) \tag{2}$$

To approximate the score-function they train a denoising function $D_\theta(\mathbf{x}(t), t)$, parameterized by $\theta$, by minimizing the loss

$$\mathcal{L}_{EDM}(\theta) = \mathbb{E}\left[\lambda(t) \, \|D_\theta(\mathbf{x}(t), t) - \mathbf{x}(0)\|_2^2\right] \tag{3}$$

where the expectation is over $\mathbf{x}(0) \sim p_{data}$, $\ln(t) \sim \mathcal{N}(P_{mean}, P_{std}^2)$ and the noise added to $\mathbf{x}(0)$ to get $\mathbf{x}(t)$, $\epsilon \sim \mathcal{N}(\mathbf{0}, \mathbf{I})$. $P_{mean}$ and $P_{std}$ are hyperparameters. Given a trained denoising function the score function can be retrieved with

$$\nabla_{\mathbf{x}(t)} \log p(\mathbf{x}(t), t) = \frac{D_\theta(\mathbf{x}(t), t) - \mathbf{x}}{t^2} \tag{4}$$

Furthermore, they employ a concept they call pre-conditioning, meaning the denoising function is given by

$$D_\theta(\mathbf{x(t)}, t) = c_{skip}(t) \, \mathbf{x}(t) + c_{out}(t) \, F_\theta(c_{in}(t) \, \mathbf{x}(t), c_{noise}(t)) \tag{5}$$

where $F_\theta$ is a function represented by a neural network. The mathematical expressions for the scalars $c_{skip}(t)$, $c_{out}(t)$ and $c_{in}(t)$ are derived from the requirements that the input and outputs vectors of the network should have unit variance, and to amplify errors made by $F_\theta$ as little as possible. The weight $\lambda(t)$ in the loss (Equation 3) is chosen so that the loss for individual time steps are weighted equally as viewed by the network $F_\theta$. Lastly $c_{noise}$ is chosen empirically.

## 3.2 EDM2 AND TOOLS FOR STANDARDIZATION

Several recent works have observed large magnitudes in various intermediate representations inside Deep Neural Networks (Polyak et al., 2024; Darcet et al., 2023; Karras et al., 2024). Most relevant to this work is the paper by Karras et al. (2024), who report that both the networks weights and intermediate activations can grow uncontrollably when training a diffusion-based model using the ADM architecture (Dhariwal & Nichol, 2021). This potentially leaves the network in a constant unconverged state.

To handle this uncontrollable growth Karras et al. (2024) propose significant architecture changes to the network. A central concept introduced to make these changes is *Expected magnitude*, defined as

$$\mathcal{M}[\mathbf{a}] = \sqrt{\frac{1}{N^a} \sum_{i=1}^{N^a} \mathbb{E}[a_i^2]} \tag{6}$$

where $\mathbf{a}$ is a vector of dimensionality $N^a$. The vector $\mathbf{a}$ is called standardized iff $\mathcal{M}[\mathbf{a}] = 1$.

For input feature normalization purposes we can achieve standardized feature vectors in multiple ways. One is simply to normalize our data to have zero mean and unit variance, meaning that regular z-score normalization achieves standardization.

A standardized operation is then defined as an operation when given a standardized vector as input, outputs a standardized vector. Karras et al. (2024) define several standardized operations, the one we make explicit use of is the magnitude-preserving concatenation operator. While concatenating two or more standardized vectors does keep the result standardized, their impact on the output depends on the their relative dimensionality. To make this explicitly controllable Karras et al. (2024) define the following weighting scheme

$$\mathbf{c} = \sqrt{\frac{N^a + N^b}{(1-\alpha)^2 + \alpha^2}} \left[ \frac{1-\alpha}{N^a} \mathbf{a} \oplus \frac{\alpha}{N^b} \mathbf{b} \right] \tag{7}$$

where $\mathbf{a}$ and $\mathbf{b}$ are two vectors, $N^a$ and $N^b$ are their corresponding dimensionalities, $\oplus$ is the concatenation operator and $\alpha$ is a explicit parameter controlling the contribution of each vector on the concatenated vector $\mathbf{c}$ while keeping $\mathcal{M}[\mathbf{c}] = 1$.

Lastly, to standardize the weighting of the loss as training progresses they propose a continuous generalization of the uncertainty based task weighting first proposed by Kendall et al. (2018)

$$\mathcal{L}_{EDM2}(\theta, \psi) = \mathbb{E}\left[ \frac{\mathcal{L}_{EDM}(\theta)}{e^{u_\psi(t)}} + u_\psi(t) \right] \tag{8}$$

where $u_\psi(t)$ is a Fourier features (Tancik et al., 2020) embedding of the timestep and a one layer MLP (represented with $\psi$), which is trained jointly with $D_\theta$ using the same loss[1].

## 4 STEPS TOWARDS PRINCIPLED HUMAN MOTION AND SHAPE GENERATION

In this section, we detail the adaptations made for human motion and shape generation. We start by describing our SMPL-based parameterization of human pose and motion, followed by the baseline training setup. We then introduce a series of targeted modifications, each motivated to address a specific imbalance during training and evaluated for its impact on performance (see Table 1). Descriptions of the evaluation protocol are provided in Section 5 and Appendix A.

### 4.1 MOTION REPRESENTATION

We represent human motion with the SMPL (Loper et al., 2023) parameters. A motion is represented as $\mathbf{x}(0) \in \mathbb{R}^{N \times L}$ where $L = 192$ is the sequence length and $N$ the dimensionality of the feature

---

[1]There is a small discrepancy in Equation 8 which is adopted from the supplementary material of (Karras et al., 2024). In practice $u_\psi(t)$ is applied before sum reduction, see `https://github.com/NVlabs/edm2/blob/main/training/training_loop.py`. Meaning the equation should be ... $+ Mu(t)$, where $M$ is the number of elements in a sample. However as we use mean reduction the $M$ will cancel out again, and because a scalar scaling of the loss will have no impact when using the Adam optimizer, we leave it as is.

Table 1: Collection of results from each ablation performed in Section 4. Ablations and results are cumulative, meaning a section's experiment also includes changes from all previous sections. Each of our proposed additions improve the model. FID is calculated on the validation set.

| Ablations | FID ↓ | Diversity ↑ | Foot skating (%) ↓ | Limb $\sigma$ (mm) ↓ |
|---|---|---|---|---|
| 4.2 Baseline | 6.23 | 6.59 | 44.33 | 0.15 |
| 4.3 Input feature normalization | 3.32 | 7.84 | 34.84 | 0.07 |
| 4.4 Gradient analysis | 2.65 | 8.04 | 32.31 | 0.06 |
| 4.5 Per group weighting | 2.48 | 7.96 | 31.66 | 0.05 |
| 4.6 Addressing dimensionality | 2.40 | 8.00 | 20.43 | 0.02 |

vector for one frame composed of SMPL parameters. Each element in the motion sequence $\mathbf{x}_i$, referred to as a frame, is given by

$$\mathbf{x}(0)_i = \begin{bmatrix} J_i & \Phi_i & \tau_i & \beta \end{bmatrix}^T \tag{9}$$

Where $J_i \in \mathbb{R}^{N^J}$, $N^J = 21 * 6$ is the 21 SMPL joint angles representing the pose of the body, $\Phi_i \in \mathbb{R}^{N^\Phi}, N^\Phi = 6$ is the global orientation, $\tau_i \in \mathbb{R}^{N^\tau}$, $N^\tau = 3$ is the global absolute position and $\beta \in \mathbb{R}^{N^\beta}$, $N^\beta = 10$ is the shape. Rotations are represented in 6D as proposed by Zhou et al. (2019). Throughout this work we will refer to $J_i$, $\Phi_i$, $\tau_i$ and $\beta$ as groups of features, or simply group. For notational convenience we will consider the motions as $NL$ dimensional vectors.

## 4.2 BASELINE GENERATIVE METHOD

Our baseline generative method follows the EDM framework (Karras et al., 2022), including the use of the EDM2 network (Karras et al., 2022). Here, we provide the details specific to our implementation. The design choices and hyperparameters were selected based on prior work (Karras et al., 2022; 2024; Guo et al., 2022) and limited empirical tuning, to support meaningful comparisons across ablations. Key differences from the original EDM framework include a cosine decay learning rate schedule and an increased $t_{min}$, both which had a notable impact on performance. A complete description of implementation details can be found in Appendix A.

Prior to adding noise to our data samples $\mathbf{x}(0)$ we employ input feature normalization. Our baseline follows the HumanML3D (Guo et al., 2022) methodology, z-score normalization with an element wise mean, and a standard deviation which is the mean standard deviation of each feature group. We undo this as a last step after sampling. This means that the standard deviation of our data is 1.2 as viewed by the pre-conditioning (i.e. $\sigma_{data} = 1.2$ in (Karras et al., 2022)). We will denote our normalized data as $\hat{\mathbf{x}}(0)$ and the normalized data at timestep $t$ as $\hat{\mathbf{x}}(t)$

We employ the $\mathcal{L}_{EDM2}(\theta, \psi)$ loss (Equation 8) with some modifications:

$$\mathcal{L}(\theta, \psi) = \mathbb{E} \left[ \frac{\lambda(t)}{NLe^{u_\psi(t)}} \| D_\theta(\hat{\mathbf{x}}(t), t) - \hat{\mathbf{x}}(0) \|_2^2 + u_\psi(t) \right] \tag{10}$$

where the expectation is taken over $\hat{\mathbf{x}}(0) \sim \hat{p}_{data}$, $\ln(t) \sim \mathcal{N}(P_{mean}, P_{std}^2)$ and $\epsilon \sim \mathcal{N}(\mathbf{0}, \mathbf{I})$. The distribution of our feature normalized motion sequences is represented by $\hat{p}_{data}$. Note how Equation 10 assumes equal length sequences which is not the case in practice. We will continue with this assumption throughout this paper, please see Appendix A for more details on how training with variable length sequences is handled in practice.

## 4.3 INPUT FEATURE NORMALIZATION USING EXPECTED MAGNITUDE

The input feature normalization employed in HumanML3D (Guo et al., 2022) and our baseline, combined with the pre-conditioning in the EDM framework (Karras et al., 2022) ensures that our neural network inputs and outputs are standardized. Yet we can do better.

There are two problems with the current input feature normalization. First, it does not lead to equal variance between the feature groups, which we hypothesize leads to imbalances in the training dynamics. Secondly, the structure of our rotation features are not preserved, making it harder to learn.

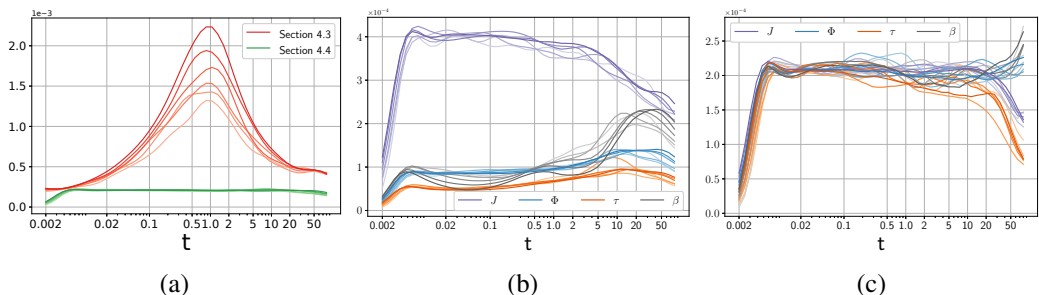

(a)                          (b)                          (c)

Figure 2: Average L2 norms of gradients with respect to $F_\theta$ over diffusion time steps at different points during training. From epoch 100 (lightest) to epoch 600 (darkest). Calculated using PyTorch autograd, with a batch size of 32 and averaged over entire training dataset. **(a)** Gradient norms *before and after* re-balancing. **(b)** Gradient norms per feature group *before* per feature group balancing. **(c)** Gradient norms per feature group *after* per feature group balancing.

Consider the rotations $R$ in the pose $J$ and global orientation $\Phi$. Subtracting element wise mean destroys the orthogonality of the column vectors $r_i$ needed to represent each 3D rotation $R$. Instead of using statistics to standardize, we can use the fact that $r_i$ are unit vectors with expected magnitude

$$\mathcal{M}[r_i] = \|r_i\|_2/\sqrt{3} = \frac{1}{\sqrt{3}} \tag{11}$$

To standardize them we can simply multiply with $\sqrt{3}$. However, since our rotation features would not be zero-mean anymore, this breaks the original derivation of the pre-conditioning, based on scaling to unit variance (Karras et al., 2022). Interestingly though, the same exact pre-conditioning also holds for scaling to standardization if we replace the standard deviation of the data with the expected magnitude of the data (i.e. $\sigma_{data} = \mathcal{M}[\mathbf{x}(0)_i] = 1$). Removing the requirement of zero mean features. Please refer to Appendix B for details.

For the global translation $\tau$ we want to avoid skewing the 3D space. Or in other words we want to avoid scaling each coordinate differently. We employ z-score normalization with mean and standard deviation calculated over all coordinates.

Although the SMPL model defines $\beta$ to have zero mean and unit variance, the training data we use (Section 5.1) does not strictly follow this distribution in practice. Since each element in $\beta$ represents a weight for a Principal Component direction, we can safely do elementwise z-score normalization.

After these changes to the input feature normalization, each feature group is individually standardized which is also the case for the full feature vector containing all feature groups.

## 4.4 GRADIENT ANALYSIS OF UNCERTAINTY WEIGHTING

Due to network initialization, input feature normalization, pre-conditioning and $\lambda(t)$, the original EDM loss $\mathcal{L}_{EDM}(\theta)$(Karras et al., 2022) (Equation 3) is equally weighted over different time steps at the start of training. However, this is not necessarily the case as training progresses. Thus Karras et al. (2024) proposed a continuous generalization of the uncertainty based task weighting first proposed by Kendall et al. (2018), resulting in the loss adopted by our work (referring to Equation 8 and Equation 10). The goal of these losses are to adaptively balance the loss between time steps as training progresses. The intuition provided by Karras et al. (2024) is based on setting the derivative of the loss with respect $u_\psi(t)$ to 0 and solving for $e^{u_\psi(t)}$

$$\frac{d}{du_\psi(t)}\mathcal{L}(\theta, \psi) = 0 \quad \Rightarrow \quad e^{u_\psi^*(t)} = \mathbb{E}\left[\frac{\lambda(t)}{NL}\|D_\theta(\hat{\mathbf{x}}(t), t) - \hat{\mathbf{x}}(0)\|_2^2\right] \tag{12}$$

where $u_\psi^*(t)$ is the optimal prediction. The idea being that, if $u_\psi(t)$ has converged to the optimal prediction, the loss is divided by its reciprocal, equalizing the loss contribution over time steps. However, while equalizing the loss, this is not the case for the gradients. Looking at the expected magnitude of the gradient of $\mathcal{L}(\theta, \psi)$ with respect to $F_\theta$ (if we substitute $e^{u_\psi^*(t)}$ for the RHS in

Equation 12) we can see that it is proportional to reciprocal of the square root of the L2 norm, which grows as the loss decreases, and does not guarantee that the gradient is equalized over time steps.

$$\mathcal{M}[\nabla_{F_\theta}\mathcal{L}(\theta, \psi)] \propto \frac{c_{out}(t)}{\sqrt{\mathbb{E}[\|D_\theta(\hat{\mathbf{x}}(t), t) - \hat{\mathbf{x}}(0)\|_2^2]}} \tag{13}$$

Instead we consider the expected magnitude of the gradients of the unweighted loss

$$\mathcal{M}\left[\nabla_{F_\theta}\mathbb{E}\left[\frac{1}{NL}\|D_\theta(\hat{\mathbf{x}}(t), t) - \hat{\mathbf{x}}(0)\|_2^2\right]\right] \propto c_{out}(t)\sqrt{\frac{1}{NL}\mathbb{E}[\|D_\theta(\hat{\mathbf{x}}(t), t) - \hat{\mathbf{x}}(0)\|_2^2]} \tag{14}$$

Which is proportional to the square root of the unweighted loss, times $c_{out}(t)$. Even though the uncertainty based weighting results in an unsatisfactory weight, it is a convenient way to learn the expression inside the square root in Equation 14. To achieve this affect, we can set up a separate loss for training $u_\psi(t)$ using the stop gradient operator $\oslash$ (.detach() in PyTorch)

$$\mathcal{L}(\psi) = \mathbb{E}\left[\frac{1}{NLe^{u_\psi(t)}} \oslash (\|D_\theta(\hat{\mathbf{x}}(t), t) - \hat{\mathbf{x}}(0)\|_2^2) + u_\psi(t)\right] \tag{15}$$

Now we can use $\sqrt{e^{u_\psi(t)}}$ as an approximation for the square root in Equation 14 which we can use to scale our loss. We also need to scale by $\frac{1}{c_{out}(t)}$. This is incidentally equal to $\sqrt{\lambda(t)}$. Our denoising function loss becomes

$$\mathcal{L}(\theta) = \mathbb{E}\left[\frac{\sqrt{\lambda(t)}}{NL \oslash (\sqrt{e^{u_\psi(t)}})}\|D_\theta(\hat{\mathbf{x}}(t), t) - \hat{\mathbf{x}}(0)\|_2^2\right] \tag{16}$$

While this method will equalize the losses over $t$ it does not standardize them. However, we do not care about the absolute expected magnitude as this has no impact on the optimum. Figure 2a depicts how the gradients behave in practice before and after the change proposed in this section. Please refer to Appendix C for detailed derivations. As final practical note, we noticed that $u_\psi(t)$ was not expressive enough to converge to the optimum. To address this, we added a learnable gain, similar to the way it's used in the last layer of the main EDM2 network. (Karras et al., 2024).

### 4.5 PER FEATURE GROUP UNCERTAINTY WEIGHTING

With the proposed modification described in the last section, our loss is now properly weighted over different $t$. However, one issue remains. We use a single weight, proportional to the expectation over all features. In practice, the average loss can differ between feature groups. To handle this, with a slight abuse of notation, we define a $u_\psi(t)$ for each feature group

$$u_\psi(t) = \begin{bmatrix} u_{\psi^J}^J(t) & u_{\psi^\Phi}^\Phi(t) & u_{\psi^\tau}^\tau(t) & u_{\psi^\beta}^\beta(t) \end{bmatrix}^T \tag{17}$$

which we can apply separately to each group. To save space we do not re-state the loss in this section, please refer to Appendix D for details. In practice we predict each part of $u_\psi(t)$ with a separate MLP, each architecture as described previously. The overhead of $u_\psi(t)$ is still low, the networks are very lightweight, each one takes about 1 ms per batch in total for a forward and backward pass on a NVIDIA RTX 3090 GPU.

### 4.6 ADDRESSING FEATURE GROUP DIMENSIONALITY

Due to our input feature normalization and the input scaling $c_{in}(t)$ the inputs to our network are standardized i.e. $\mathcal{M}[c_{in}(t)\mathbf{x}(t)] = 1$. However, the contribution of each feature group to $\mathbf{x}(0)$ is proportional to its dimensionality (Karras et al., 2024). The situation is similar for our loss, the norm of the gradient with respect to each group is proportional to the groups dimensionality. Our previous loss balancing looks at expected magnitude, which divides with feature group size, effectively balancing the individual gradient elements. While it maybe would have been possible to construct a loss such that $e^{u_\psi^*(t)}$ accounts for feature group dimensionality, we know the exact impact of the dimensionality and do not have to use a learned approximation to account for it. In both cases,

we can use the magnitude preserving concatenation operator. We can generalize Equation 7 to four equally weighted vectors and write it as a weight for each feature group

$$w^k = \sqrt{\frac{N^J + N^\Phi + N^\tau + N^\beta}{4}} \frac{1}{\sqrt{N^k}} \tag{18}$$

where k is either $J$, $\Phi$, $\tau$ or $\beta$. For the inputs, we apply the weight after adding noise. We define a frame of the weighted inputs $\hat{\mathbf{x}}_w(t)_i$ as

$$\hat{\mathbf{x}}_w(t)_i = \begin{bmatrix} w^J \hat{\mathbf{x}}^J(t)_i & w^\Phi \hat{\mathbf{x}}^\Phi(t)_i & w^\tau \hat{\mathbf{x}}^\tau(t)_i & w^\beta \hat{\mathbf{x}}^\beta(t)_i \end{bmatrix}^T \tag{19}$$

With the weights applied, our final loss is

$$\mathcal{L}_{\text{final}}(\theta) = \sum_{k \in \{J, \Phi, \tau, \beta\}} \mathbb{E} \left[ \frac{\sqrt{\lambda(t)} w^k}{NL \oslash (\sqrt{e^{u^k_{\psi^k}(t)}})} \| D^k_\theta(\hat{\mathbf{x}}_w(t), t) - \hat{\mathbf{x}}^k(0) \|^2_2 \right] \tag{20}$$

See Figure 2b and Figure 2c on how the per feature group gradients behave before and after the changes described in this and previous section. Note how the gradients are less well balanced in regions where few diffusion time steps $t$ are sampled. However, because these regions are sampled less frequently, their impact on overall training remains limited. After the changes described in this section, we arrive at our final model.

Table 2: Quantitative comparison between our final models and two other generative human motion diffusion models. FID is calculated on the test set. Best in each column is **bold**, second best is underlined. The Real row depicts metrics calculated on training data.

| Methods | NFE | FID ↓ | Diversity ↑ | Foot skating (%) ↓ | Limb $\sigma$ (mm) ↓ |
|---|---|---|---|---|---|
| MDM | 1000 | 3.58 | 8.14 | 8.58 | 3.73 |
| MLD | 50 | **1.17** | 8.20 | 18.99 | 5.89 |
| Ours[Root rel.] | 31 | 3.18 | **8.76** | **7.97** | 1.74 |
| Ours[SMPL] | 31 | 1.81 | 8.73 | 16.31 | **0.02** |
| Real | - | 0.06 | 9.56 | 6.32 | 5.2e-5 |

## 5 EXPERIMENTS

### 5.1 DATASET AND EVALUATION METRICS

We use the portion of the AMASS dataset (Mahmood et al., 2019) that is included in HumanML3D (Guo et al., 2022). This choice provides full SMPL parameters and is directly compatible with prior works (Chen et al., 2023; Tevet et al., 2023).

Following standard practice in human motion generation, we report Fréchet Inception Distance (FID) and Diversity (Guo et al., 2022; Chen et al., 2023; Tevet et al., 2023). We complement these with two additional quality metrics. Foot skating (Zhang et al., 2024b), which reflects foot-contact quality, and limb-length standard deviation (Limb $\sigma$), which measures how consistent limb lengths remain over time. See Appendix A for details.

### 5.2 COMPARISON TO PREVIOUS WORKS

We compare against two human motion diffusion models, MDM (Tevet et al., 2023) and MLD (Chen et al., 2023). Both use the HumanML3D (Guo et al., 2022) input-feature parameterization, MLD applies diffusion in a VAE latent space derived from these features. For MLD we evaluate a pre-trained model provided by the authors. MDM reports its unconditional evaluation on the HumanAct12 dataset (Guo et al., 2020), to enable a direct comparison on HumanML3D, we re-train an unconditional MDM dataset using the authors' released code and default hyperparameters.

Neither prior work directly predicts shape, and the metrics are computed either in the HumanML3D feature space or from 3D joint coordinates. To enable a more nuanced comparison, in addition to

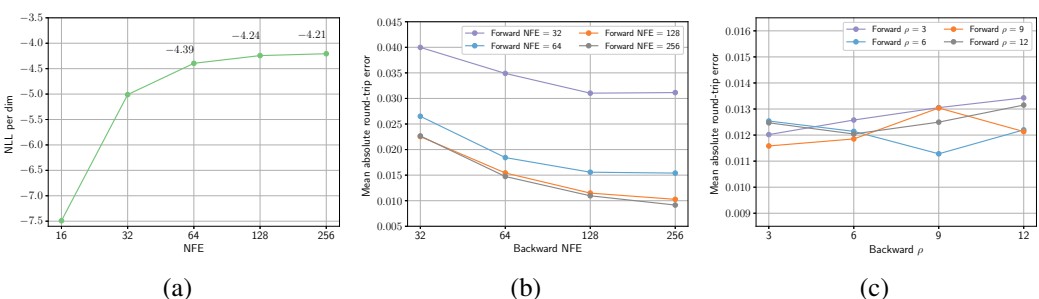

(a)           (b)           (c)

Figure 3: PF-ODE evaluation on the full-length test motions. **(a)** NLL in unnormalized feature space vs. NFE. **(b)** Round-trip error in normalized feature space vs. backwards NFE. **(c)** Round trip error in normalized feature space vs. backwards $\rho$.

our SMPL-parameterized model we also train a model on root-relative motion features. For the final results, we increase both training time and model size relative to the ablations (see Appendix A).

Table 2 summarizes the results. Our models are on par with prior works overall while requiring fewer sampling steps. Notably, methods that diffuse in a feature space without 3D joint coordinates (Ours[SMPL] and MLD), achieve the best FID. In contrast, methods that diffuse in a feature space with 3D joint coordinates (Ours[Root rel.] and MDM) obtain better foot-contact as measured by foot skating. However, these parameterization incur time-varying limb lengths, reflected by the Limb $\sigma$ metric, and the need for slow post-processing to infer shape. Finally, our models achieve the best diversity and lowest Limb $\sigma$ across parameterizations.

We encourage readers to view the videos in the supplementary material for a quantitative comparison.

### 5.3 PROBABILITY FLOW ODE EXPERIMENTS

Two key advantages of the PF-ODE are tractable likelihoods and convergence of ODE trajectories under reasonable solvers and schedules. We evaluate both on all full-length test motions, results are in Figure 3. For likelihoods, the average log-likelihood in the unnormalized feature space plateaus by 128–256 NFE. To quantify ODE trajectory convergence, we report round trip error (RTE) in the normalized feature space. Starting from a test sample, we integrate forward along the PF-ODE to the prior distribution, then integrate back to reconstruct the sample and measure mean absolute error. We sweep combinations of forward and backward discretizations. We observe no adverse effect from asymmetric discretizations. Holding one side fixed and adding NFE on the other reliably decreases RTE. Likewise, matching $\rho$ (discretization parameter per Karras et al. (2022)) between forward and backward ODE solves offers no advantage. See Appendix A for implementation details. In the supplementary videos, we also visualize the top ten and bottom ten full-length test samples ranked by our model's likelihood.

## 6 FUTURE WORK

**Conditional generation** Our proposed method is well suited as a base model on which conditional human motion and shape generation can be built. Importantly, nothing in the underlying framework needs to change in the conditional case. The standard score-matching loss, the motion features at diffusion timestep $t$ and their weightings remain the same. The only requirement is that the conditioning signal is normalized to have unit expected magnitude. This covers a wide range of practical conditioning setups, such as text descriptions (text-to-motion), action labels (action-to-motion), and partially observed or occluded motions (infilling) as well as common techniques such as classifier-free guidance Ho & Salimans (2022).

**Beyond human motion generative models** We believe the principles underlying our approach could be applied to other human pose, motion, and shape models, or even entirely different domains where inputs or targets consist of heterogeneous feature types. This opens several promising directions for future work.

## 7 CONCLUSION

In this work, we have demonstrated that achieving performance on par with state-of-the-art methods in unconditional human motion and shape generation is possible using score-based diffusion models, without relying on auxiliary regularization losses or redundant representations of human motion during training. By adapting and extending tools originally developed by Karras et al. (2024), our approach consists of theoretically motivated weighting of the standard score-matching L2 loss, combined with careful normalization of feature groups within a minimal SMPL-based motion representation. We have individually assessed the effectiveness of each proposed component, highlighting their contributions to the overall performance of the model.

## 8 REPRODUCIBILITY STATEMENT

While we believe our main text gives an good idea on how our method works, much more detail is added in the Appendix. Appendix A contains implementation details about data handling, evaluation, network, training, handling variable length inputs, root relative parameterization and hyperparameters. Appendix B-D contains proofs. We share the network code that we have used.

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

## A  IMPLEMENTATION DETAILS

In this section we will describe implementation details that we did not have room for in the main paper. See Table 3 for all hyperaparameters.

### A.1  DATA HANDLING DETAILS

We crop all motions to the maximum length of $L = 192$ frames sequences, and if shorter we crop them to be divisible by 16. Sequences shorter than $L$ are zero padded to $L$. We drop all motions shorter than 32 frames. Resulting in 10626 training, 665 validation and 1997 testing samples. All motions are put in a canonical space such that each sequence starts at $(0, 0, z)$ and the ground plane is at $(x, y, 0)$ which we assume to be flat with no incline as is common (Zhang et al., 2024b; Guo et al., 2022; Chen et al., 2023; Tevet et al., 2023). In contrast to previous work (Zhang et al., 2024b; Guo et al., 2022; Chen et al., 2023; Tevet et al., 2023) we do not change the global orientation to face a certain direction at the first frame. To avoid excessive architectural changes, we copy the same shape vector $\beta$ to the representation of each frame which doesn't seem to have a significant negative impact on the end results (see Limb $\sigma$ in Table 1 and Table 2 in the main paper).

### A.2  NETWORK DETAILS

We employ the *EDM2* network (Karras et al., 2024), a U-Net. Since our data is a 1D sequence rather than 2D, we replace all 2D operations with corresponding 1D operations.

Furthermore, to handle the variable length inputs, in addition to zero padding and loss masking (see Appendix A.4), we use masking in two more places. On the inputs of every convolution layer with a $3 \times 1$ kernel except the first (inputs are already zero at padded positions and noise is only added to valid positions), ensuring the same border conditions on all data samples. In the attention layers, ensuring no attending to padded positions.

We supply a python file (edm2.py) containing the PyTorch implementation of the network in the supplementary material.

Table 3: Hyperparameters used for all versions of our model.

| General hyperparameter | Ablations | Final (SMPL) | Final (Root rel.) |
|---|---|---|---|
| Batch size | 64 | 64 | 64 |
| Training epochs | 600 | 2100 | 1500 |
| Max learning rate | 1e-2 | 1e-2 | 1e-2 |
| Warm up epochs | 10 | 10 | 10 |
| Adam $\beta_1, \beta_2$ | 0.9, 0.95 | 0.9, 0.95 | 0.9, 0.95 |
| $P_{mean}$ | -1.2 | -1.2 | -1.2 |
| $P_{std}$ | 1.2 | 1.2 | 1.2 |
| **Model hyperparameter** | | | |
| Base channels | 192 | 192 | 192 |
| Channel multipliers | 1, 2, 3, 4 | 1, 2, 3, 4 | 1, 2, 3, 4 |
| Blocks per resolution | 1 | 3 | 3 |
| Attn. resolutions | $\frac{1}{4}, \frac{1}{8}$ | $\frac{1}{4}, \frac{1}{8}$ | $\frac{1}{4}, \frac{1}{8}$ |
| Dropout | 0 | 0.1 | 0.1 |
| **Sampling hyperparameter** | | | |
| ODE solver | Heun 2nd | Heun 2nd | Heun 2nd |
| $t_{min}$ | 0.02 | 0.02 | 0.02 |
| $\rho$ | 9 | 9 | 9 |
| NFE | 31 | 31 | 31 |

## A.3 TRAINING DETAILS

We use the Adam optimizer (Kingma & Ba, 2014) with a cosine decay to zero learning rate schedule (Kingma & Ba, 2014; Karras et al., 2024) and a linear warm up. We use two data augmentations. Rotation around the up-axis with a uniformly sampled angle, as well as random (with a 50% probability) mirroring of the right and left limbs.

## A.4 LOSS FOR VARIABLE LENGTH

For all the losses in the main paper and the Appendix we assume equal length motions, where the length is $L = 192$. However, in practice the inputs are of variable length. To explain how we deal with this, we will consider our baseline loss (Equation 10 in the main paper), but it applies to all losses. Our baseline loss is re-stated for convenience:

$$\mathcal{L}(\theta, \psi) = \mathbb{E}\left[ \frac{\lambda(t)}{NLe^{u_\psi(t)}} \|D_\theta(\hat{\mathbf{x}}(t), t) - \hat{\mathbf{x}}(0)\|_2^2 + u_\psi(t) \right] \tag{21}$$

where the expectation is taken over $\hat{\mathbf{x}}(0) \sim \hat{p}_{data}$, $\ln(t) \sim \mathcal{N}(P_{mean}, P_{std}^2)$ and $\epsilon \sim \mathcal{N}(\mathbf{0}, \mathbf{I})$. Mathematically, it would be wrong to consider $L$ representing the length of each variable length input, as it would indicate that we divide each sample with a different value. In practice, this is done over batches.

```
1 loss = (lambda_t / torch.exp(u_t)) * (D_theta_x_t - x_0) ** 2 + u_t
2 expected_loss = torch.sum(loss * mask) / (torch.sum(mask) * N)
```
Listing 1: PyTorch pseudo code of the loss in Equation 22

Listing 1 contains the simple PyTorch python pseudo code used to calculate our loss in practice. Mathematically we view this as an expectation over non-padded frames (or frames with valid indexes) in contrast to expectations over full motions. With some abuse of notation we can write this like

$$\mathcal{L}(\theta, \psi) = \mathbb{E}\left[ \frac{\lambda(t)}{Ne^{u_\psi(t)}} \|D_\theta(\hat{\mathbf{x}}(t), t)_i - \hat{\mathbf{x}}(0)_i\|_2^2 + u_\psi(t) \right] \tag{22}$$

where the expectation is additionally taken over $i \in \mathcal{V}$, where $\mathcal{V}$ is the set of indexes of valid frames.

### A.5 DETAILS ON ROOT-RELATIVE PARAMETERIZATIONS

Our root-relative motion representation has two feature groups, 3D joint coordinates expressed relative to the SMPL pelvis joint, and the global pelvis trajectory. We process the pelvis trajectory exactly as the SMPL global translation in our SMPL-parameterized model. It is mapped to the same canonical space and normalized in the same way. The root-relative 3D joint coordinates are z-normalized using the group-wise mean and standard deviation (computed over the training set). The procedures described in Sections 4.4–4.6 are applied identically to this representation, the only difference is that there are two feature groups instead of four.

### A.6 EVALUATION DETAILS

For FID and diversity we use the networks and definitions used by Guo et al. (2022). For FID we compare with validation and test data during ablations and testing respectively. For foot skating we follow the work by Zhang et al. (2024b).

For limb length standard deviation (Limb $\sigma$), we define a limb length as the distance between a joint and its parent in the SMPL (Loper et al., 2023) kinematic tree. We then calculate the standard deviation per limb and sequence, and average over all limbs in all generated sequences. Foot skating and limb length standard deviation are calculated on 3D joint coordinates. We use the SMPL-H (Romero et al., 2017) neutral model to convert SMPL parameters to 3D joint coordinates. For other methods we extract 3D joint coordinates following the original works. FID and diversity requires the HumanML3D motion representation, which is computed from 3D joint coordinates.

We solely generate 192 frame sequences for evaluation. We employ three runs, where we generate 5000 sequences. The best metrics over the three runs are reported. The variation is 3% or less between runs over all metrics. We checkpoint the 10 last epochs, and choose the best one according to validation set FID.

Unless otherwise stated, all results use the second-order Heun sampler from EDM (Karras et al., 2022) with 31 NFE, $\rho = 9$, and $t_{\min} = 0.02$.

### A.7 PF-ODE EXPERIMENTAL DETAILS

We calculate likelihoods with the change of variables formula (Chen et al., 2018), largely following the methodology described and implemented by Song et al. (2020b), using a Skilling-Hutchinson trace estimator (Skilling, 1989; Hutchinson, 1989) with Rademacher noise. However, instead of using a black-box RK45 ODE solver (Dormand & Prince, 1980), we utilize the second order Heun sampler from EDM Karras et al. (2022). Crucially for a both the NLL and RTE experiments we don't use a fallback to Euler when $t_{next} = 0$. Instead the lowest noise level we evaluate the PF-ODE drift at is $\epsilon = 1e - 5$. During RTE experiments, while sweeping NFE we keep $\rho = 9$, and while sweeping $\rho$ we keep NFE fixed at 128.

## B DERIVATION OF PRE-CONDITIONING

The EDM pre-conditioning (Karras et al., 2022) (here repeated in Equation 23 for convenience) and the loss weight $\lambda(t)$ was derived from first principles with several goals related to variances in mind. A prerequisite for it to work is that the data is zero mean. However, as we will show here, it turns out that the exact same pre-conditioning can be used if we re-frame the goals in terms of expected magnitude.

$$D_\theta(\mathbf{x}(\mathbf{t}), t) = c_{skip}(t)\, \mathbf{x}(t) + c_{out}(t)\, F_\theta(c_{in}(t)\mathbf{x}(t), c_{noise}(t)) \tag{23}$$

The original goal of $c_{in}(t)$ was to ensure the input to the neural network $F_\theta$ is unit variance (Karras et al., 2022).

$$Var[c_{in}(t)\hat{\mathbf{x}}(t)] = 1 \tag{24}$$

$$Var[c_{in}(t)(\hat{\mathbf{x}}(0) + t\epsilon)] = 1 \tag{25}$$

$$c_{in}(t)^2 Var[\hat{\mathbf{x}}(0) + t\epsilon] = 1 \tag{26}$$

$$c_{in}(t)^2(\sigma_{data}^2 + t^2) = 1 \tag{27}$$

$$c_{in}(t) = \frac{1}{\sqrt{\sigma_{data}^2 + t^2}} \tag{28}$$

This leads to the expression in Equation 28 (Karras et al., 2022) where $\sigma_{data}$ is the standard deviation of the data. We can do a similar derivation, but by only considering expected magnitude.

$$\mathcal{M}[c_{in}(t)\hat{\mathbf{x}}(t)]^2 = 1 \tag{29}$$

$$c_{in}(t)^2 \mathcal{M}[\hat{\mathbf{x}}(t)]^2 = 1 \tag{30}$$

$$c_{in}(t)^2 \left( \frac{1}{NL} \sum_{i=1}^{NL} \mathbb{E}[\hat{\mathbf{x}}(t)_i^2] \right) = 1 \tag{31}$$

$$c_{in}(t)^2 \left( \frac{1}{NL} \sum_{i=1}^{NL} \mathbb{E}[(\hat{\mathbf{x}}(0) + t\epsilon)_i^2] \right) = 1 \tag{32}$$

$$c_{in}(t)^2 \left( \frac{1}{NL} \sum_{i=1}^{NL} \mathbb{E}[\hat{\mathbf{x}}(0)_i^2] + \mathbb{E}[t^2 \epsilon_i^2] + \underbrace{2\mathbb{E}[\hat{\mathbf{x}}(0)_i t\epsilon_i]}_{=\mathbf{0}} \right) = 1 \tag{33}$$

$$c_{in}(t)^2 \left( \mathcal{M}[\hat{\mathbf{x}}(0)]^2 + \mathcal{M}[t\epsilon]^2 \right) = 1 \tag{34}$$

$$c_{in}(t)^2 \left( \mathcal{M}[\hat{\mathbf{x}}(0)]^2 + t^2 \right) = 1 \tag{35}$$

$$c_{in}(t) = \frac{1}{\sqrt{\mathcal{M}[\hat{\mathbf{x}}(0)]^2 + t^2}} \tag{36}$$

This leads to the same expression as Equation 28 if we substitute $\sigma_{data}$ for $\mathcal{M}[\hat{\mathbf{x}}(0)]$.

The original goal of $c_{out}(t)$ was to ensure unit variance of the effective target

$$F_{target} = \frac{1}{c_{out}(t)}(\hat{\mathbf{x}}(0) - c_{skip}(t)(\hat{\mathbf{x}}(0) + t\epsilon)) \tag{37}$$

The effective target is derived by re-writing $\mathcal{L}_{EDM}$ (Equation 3) in terms of the neural network $F_\theta$ rather than the denoising function $D_\theta$. The original derivation is (Karras et al., 2022)

$$Var[\frac{1}{c_{out}(t)}(\hat{\mathbf{x}}(0) - c_{skip}(t)(\hat{\mathbf{x}}(0) + t\epsilon)] = 1 \tag{38}$$

$$\frac{1}{c_{out}(t)^2} Var[(\hat{\mathbf{x}}(0) - c_{skip}(t)(\hat{\mathbf{x}}(0) + t\epsilon)] = 1 \tag{39}$$

$$c_{out}(t)^2 = Var[(\hat{\mathbf{x}}(0) - c_{skip}(t)(\hat{\mathbf{x}}(0) + t\epsilon)] \tag{40}$$

$$c_{out}(t)^2 = Var[(1 - c_{skip}(t))\hat{\mathbf{x}}(0) + c_{skip}(t)t\epsilon] \tag{41}$$

$$c_{out}(t)^2 = (1 - c_{skip}(t))^2 \sigma_{data}^2 + c_{skip}(t)^2 t^2 \tag{42}$$

Now for the derivation using expected magnitude

$$\mathcal{M}[\frac{1}{c_{out}(t)}(\hat{\mathbf{x}}(0) - c_{skip}(t)(\hat{\mathbf{x}}(0) + t\epsilon)]^2 = 1 \tag{43}$$

$$\frac{1}{c_{out}(t)^2}\mathcal{M}[(\hat{\mathbf{x}}(0) - c_{skip}(t)(\hat{\mathbf{x}}(0) + t\epsilon)]^2 = 1 \tag{44}$$

$$c_{out}(t)^2 = \mathcal{M}[(\hat{\mathbf{x}}(0) - c_{skip}(t)(\hat{\mathbf{x}}(0) + t\epsilon)]^2 \tag{45}$$

$$c_{out}(t)^2 = \mathcal{M}[(1 - c_{skip}(t))\hat{\mathbf{x}}(0) + c_{skip}(t)t\epsilon]^2 \tag{46}$$

$$c_{out}(t)^2 = \frac{1}{NL}\sum_{i=1}^{NL}\mathbb{E}[((1 - c_{skip}(t))\hat{\mathbf{x}}(0)_i + c_{skip}(t)t\epsilon_i)^2] \tag{47}$$

$$c_{out}(t)^2 = \frac{1}{NL}\sum_{j=1}^{NL}\mathbb{E}[(1 - c_{skip}(t))^2\hat{\mathbf{x}}(0)_i^2] + \mathbb{E}[c_{skip}(t)^2t^2\epsilon_i^2] + \underbrace{2\mathbb{E}[(1 - c_{skip}(t))\hat{\mathbf{x}}(0)_i c_{skip}(t)t\epsilon_{ij}]}_{=\mathbf{0}} \tag{48}$$

$$c_{out}(t)^2 = (1 - c_{skip}(t))^2\mathcal{M}[\hat{\mathbf{x}}(0)]^2 + c_{skip}(t)^2t^2 \tag{49}$$

Again we find the same expression if we substitute $\sigma_{data}$ for $\mathcal{M}[\hat{\mathbf{x}}(0)]$. Furthermore, $c_{skip}(t)$ and $\lambda(t)$ are derived from $c_{out}(t)$ (Karras et al., 2022) and we do not have to re-derive them using expected magnitude.

## C  GRADIENT ANALYSIS OF UNCERTAINTY WEIGHTING

The intuition behind the uncertainty weighting derived by Karras et al. (2022) involves taking the derivative of $\mathcal{L}(\theta, \psi)$ with respect to $u_\psi(t)$ and solving for $e^{u_\psi(t)}$

$$0 = \frac{d}{du_\psi(t)}\mathcal{L}(\theta, \psi) \tag{50}$$

$$0 = -\mathbb{E}\left[\frac{\lambda(t)}{NLe^{u_\psi(t)}}\|D_\theta(\hat{\mathbf{x}}(t), t) - \hat{\mathbf{x}}(0)\|_2^2\right] + 1 \tag{51}$$

$$1 = \mathbb{E}\left[\frac{\lambda(t)}{NLe^{u_\psi(t)}}\|D_\theta(\hat{\mathbf{x}}(t), t) - \hat{\mathbf{x}}(0)\|_2^2\right] \tag{52}$$

$$e^{u_\psi(t)} = \mathbb{E}\left[\frac{\lambda(t)}{NL}\|D_\theta(\hat{\mathbf{x}}(t), t) - \hat{\mathbf{x}}(0)\|_2^2\right] \tag{53}$$

$$\Rightarrow e^{u_\psi^*(t)} = \mathbb{E}\left[\frac{\lambda(t)}{NL}\|D_\theta(\hat{\mathbf{x}}(t), t) - \hat{\mathbf{x}}(0)\|_2^2\right] \tag{54}$$

Our claim is that while balancing the loss over time steps $t$, it does not balance the gradients. To show why we looks at the expected magnitude of $\nabla_{F_\theta}\mathcal{L}(\theta, \psi)$. Specifically, we look at the gradients as calculated in practice (where the expectation is approximated as a mean over the batch). For one sample in the batch this becomes

$$\nabla_{F_\theta}\mathcal{L}(\theta, \psi) \approx \nabla_{F_\theta}\left[\frac{\lambda(t)}{BNLe^{u_\psi(t)}}\|D_\theta(\hat{\mathbf{x}}(t), t) - \hat{\mathbf{x}}(0)\|_2^2 + \frac{1}{B}u_\psi(t)\right] \tag{55}$$

$$= \frac{2\lambda(t)c_{out}(t)}{BNLe^{u_\psi(t)}}(D_\theta(\hat{\mathbf{x}}(t), t) - \hat{\mathbf{x}}(0)) \tag{56}$$

where $B$ is the batch size. Now we look at the squared expected magnitude of this gradient

$$\mathcal{M}\left[\nabla_{F_\theta}\mathcal{L}(\theta, \psi)\right]^2 \approx \frac{1}{NL}\sum_{i=1}^{NL}\mathbb{E}\left[\left(\frac{2\lambda(t)c_{out}(t)}{BNLe^{u_\psi(t)}}(D_\theta(\hat{\mathbf{x}}(t), t) - \hat{\mathbf{x}}(0))\right)^2\right] \tag{57}$$

$$= \frac{4c_{out}(t)^2}{B^2NL}\sum_{i=1}^{NL}\mathbb{E}\left[\left(\frac{\lambda(t)}{NLe^{u_\psi(t)}}(D_\theta(\hat{\mathbf{x}}(t), t) - \hat{\mathbf{x}}(0))\right)^2\right] \tag{58}$$

$$= \frac{4c_{out}(t)^2}{B^2NL}\frac{\lambda(t)^2}{N^2L^2(e^{u_\psi(t)})^2}\mathbb{E}\left[\|D_\theta(\hat{\mathbf{x}}(t), t) - \hat{\mathbf{x}}(0)\|_2^2\right] \tag{59}$$

Now substituting $e^{u_\psi(t)}$ for the RHS in Equation 54

$$= \frac{4c_{out}(t)^2}{B^2NL} \frac{1}{\mathbb{E}\left[\|D_\theta(\hat{\mathbf{x}}(t), t) - \hat{\mathbf{x}}(0)\|_2^2\right]} \tag{60}$$

Taking the square root to get the expected magnitude

$$\mathcal{M}\left[\nabla_{F_\theta}\mathcal{L}(\theta, \psi)\right] \approx \sqrt{\frac{4c_{out}(t)^2}{B^2NL} \frac{1}{\mathbb{E}\left[\|D_\theta(\hat{\mathbf{x}}(t), t) - \hat{\mathbf{x}}(0)\|_2^2\right]}} \tag{61}$$

$$\propto \frac{c_{out}(t)}{\sqrt{\mathbb{E}\left[\|D_\theta(\hat{\mathbf{x}}(t), t) - \hat{\mathbf{x}}(0)\|_2^2\right]}} \tag{62}$$

We arrive at Equation 13 from the main paper, which indicates that the gradients grow as the loss gets lower and not guaranteeing that they are balanced over time steps $t$.

Our proposed solution involves looking at the expected magnitude of the gradients of an unweighted loss. Again we start by calculating the squared expected magnitude

$$\mathcal{M}\left[\nabla_{F_\theta}\mathbb{E}\left[\frac{1}{NL}\|D_\theta(\hat{\mathbf{x}}(t), t) - \hat{\mathbf{x}}(0)\|_2^2\right]\right]^2 \approx \frac{1}{NL}\sum_{i=1}^{NL}\mathbb{E}\left[\left(\frac{2c_{out}(t)}{BNL}(D_\theta(\hat{\mathbf{x}}(t), t) - \hat{\mathbf{x}}(0))\right)^2\right] \tag{63}$$

$$= \frac{4c_{out}(t)^2}{B^2N^3L^3}\sum_{i=1}^{NL}\mathbb{E}\left[((D_\theta(\hat{\mathbf{x}}(t), t) - \hat{\mathbf{x}}(0)))^2\right] \tag{64}$$

$$= \frac{4c_{out}(t)^2}{B^2N^3L^3}\mathbb{E}\left[\|D_\theta(\hat{\mathbf{x}}(t), t) - \hat{\mathbf{x}}(0)\|_2^2\right] \tag{65}$$

$$\tag{66}$$

Taking the square root to get the expected magnitude

$$\mathcal{M}\left[\nabla_{F_\theta}\mathbb{E}\left[\frac{1}{NL}\|D_\theta(\hat{\mathbf{x}}(t), t) - \hat{\mathbf{x}}(0)\|_2^2\right]\right] \approx \sqrt{\frac{4c_{out}(t)^2}{B^2N^3L^3}\mathbb{E}\left[\|D_\theta(\hat{\mathbf{x}}(t), t) - \hat{\mathbf{x}}(0)\|_2^2\right]} \tag{67}$$

$$\propto c_{out}(t)\sqrt{\frac{1}{NL}\mathbb{E}\left[\|D_\theta(\hat{\mathbf{x}}(t), t) - \hat{\mathbf{x}}(0)\|_2^2\right]} \tag{68}$$

Thus we have arrived at Equation 14 in the main paper. To learn the expression inside the square root Equation 68 we use

$$\mathcal{L}(\psi) = \mathbb{E}\left[\frac{1}{NLe^{u_\psi(t)}} \oslash (\|D_\theta(\hat{\mathbf{x}}(t), t) - \hat{\mathbf{x}}(0)\|_2^2) + u_\psi(t)\right] \tag{69}$$

It's easy to see that this achieves the intended goal by utilizing Equations 50-54 with $\lambda(t) = 1$

## D    DERIVATION OF PER FEATURE GROUP UNCERTAINTY WEIGHTING

The only change in this section is that we apply a weighting to each feature group:

$$u_\psi(t) = \begin{bmatrix} u_{\psi^J}^J(t) & u_{\psi^\Phi}^\Phi(t) & u_{\psi^\tau}^\tau(t) & u_{\psi^\beta}^\beta(t) \end{bmatrix}^T \tag{70}$$

We can divide the squared L2 norm and write our loss for $u_\psi(t)$ as

$$\mathcal{L}(\psi) = \mathbb{E}\left[\frac{1}{NLe^{u_{\psi^J}^J(t)}} \oslash (\|D_\theta^J(\hat{\mathbf{x}}(t), t) - \hat{\mathbf{x}}^J(0)\|_2^2) + \frac{N^J L}{NL}u_{\psi^J}^J(t)\right] \tag{71}$$

$$+ \mathbb{E}\left[\frac{1}{NLe^{u_{\psi^\Phi}^\Phi(t)}} \oslash (\|D_\theta^\Phi(\hat{\mathbf{x}}(t), t) - \hat{\mathbf{x}}^\Phi(0)\|_2^2) + \frac{N^\Phi L}{NL}u_{\psi^\Phi}^\Phi(t)\right] \tag{72}$$

$$+ \mathbb{E}\left[\frac{1}{NLe^{u_{\psi^\tau}^\tau(t)}} \oslash (\|D_\theta^\tau(\hat{\mathbf{x}}(t), t) - \hat{\mathbf{x}}^\tau(0)\|_2^2) + \frac{N^\tau L}{NL}u_{\psi^\tau}^\tau(t)\right] \tag{73}$$

$$+ \mathbb{E}\left[\frac{1}{NLe^{u_{\psi^\beta}^\beta(t)}} \oslash (\|D_\theta^\beta(\hat{\mathbf{x}}(t), t) - \hat{\mathbf{x}}^\beta(0)\|_2^2) + \frac{N^\beta L}{NL}u_{\psi^\beta}^\beta(t)\right] \tag{74}$$

Using the same procedure as before we can find the optimal prediction for each feature group $k$

$$0 = \frac{d}{du^k_{\psi^k}(t)} \mathcal{L}(\psi) \tag{75}$$

$$0 = -\mathbb{E}\left[\frac{1}{NLe^{u^k_{\psi^k}(t)}} \|D^k_\theta(\hat{\mathbf{x}}(t),t) - \hat{\mathbf{x}}^k(0)\|^2_2\right] + \frac{N^kL}{NL} \tag{76}$$

$$\frac{N^kL}{NL} = \mathbb{E}\left[\frac{1}{NLe^{u^k_{\psi^k}(t)}} \|D^k_\theta(\hat{\mathbf{x}}(t),t) - \hat{\mathbf{x}}^k(0)\|^2_2\right] \tag{77}$$

$$e^{u^k_{\psi^k}(t)} = \mathbb{E}\left[\frac{1}{N^kL} \|D^k_\theta(\hat{\mathbf{x}}(t),t) - \hat{\mathbf{x}}^k(0)\|^2_2\right] \tag{78}$$

$$\Rightarrow e^{u^{*k}_{\psi^k}(t)} = \mathbb{E}\left[\frac{1}{N^kL} \|D^k_\theta(\hat{\mathbf{x}}(t),t) - \hat{\mathbf{x}}^k(0)\|^2_2\right] \tag{79}$$

To see if this is what we want, we take a look at the expected magnitude of the gradients again. This time only considering each feature group

$$\mathcal{M}\left[\nabla_{F^k_\theta} \mathbb{E}\left[\frac{1}{NL} \|D^k_\theta(\hat{\mathbf{x}}(t),t) - \hat{\mathbf{x}}^k(0)\|^2_2\right]\right]^2 \approx \frac{1}{N^kL} \sum_{i=1}^{N^kL} \mathbb{E}\left[\left(\frac{2c_{out}(t)}{BNL}(D^k_\theta(\hat{\mathbf{x}}(t),t) - \hat{\mathbf{x}}^k(0))\right)^2\right] \tag{80}$$

$$= \frac{4c_{out}(t)^2}{B^2N^kN^2L^3} \sum_{i=1}^{N^kL} \mathbb{E}\left[\left((D^k_\theta(\hat{\mathbf{x}}(t),t) - \hat{\mathbf{x}}^k(0))\right)^2\right] \tag{81}$$

$$= \frac{4c_{out}(t)^2}{B^2N^kN^2L^3} \mathbb{E}\left[\|D^k_\theta(\hat{\mathbf{x}}(t),t) - \hat{\mathbf{x}}^k(0)\|^2_2\right] \tag{82}$$

Taking the square root and we get

$$\mathcal{M}\left[\nabla_{F^k_\theta} \mathbb{E}\left[\frac{1}{NL} \|D^k_\theta(\hat{\mathbf{x}}(t),t) - \hat{\mathbf{x}}^k(0)\|^2_2\right]\right] \approx \sqrt{\frac{4c_{out}(t)^2}{B^2N^kN^2L^3} \mathbb{E}\left[\|D^k_\theta(\hat{\mathbf{x}}(t),t) - \hat{\mathbf{x}}^k(0)\|^2_2\right]} \tag{83}$$

$$\propto c_{out}(t)\sqrt{\frac{1}{N^kL} \mathbb{E}\left[\|D^k_\theta(\hat{\mathbf{x}}(t),t) - \hat{\mathbf{x}}^k(0)\|^2_2\right]} \tag{84}$$

## E    LIMITATIONS

We observe three main limitations, foot skating, self intersections and stairs. Foot skating in the SMPL parameterized model is reduced by our feature normalization and weightings, but a gap remains compared to parameterizations that include 3D joint positions. We also occasionally observe self intersections in the rendered meshes, such artifacts are present in the dataset as well. Finally, motions depicting stair walking do not always maintain the height after taking an upwards step.

## F    INFERENCE TIME

We measure the inference time of our method on an NVIDIA RTX 3090 GPU, using a batch size of 1 and averaging over 1000 samples. The measurement includes both diffusion sampling and the use of the SMPL-H (Romero et al., 2017) model to produce 3D joint positions and mesh vertices.

Diffusion sampling takes approximately $1699 \pm 71$ ms. Converting the network output to mesh vertices and 3D joint positions involves transforming from 6D rotation to axis-angle representation and running the SMPL-H model. This step adds another $31 \pm 4$ ms. No additional post-processing is applied.

In total, generating approximately 10 seconds of human motion and shape takes $1730 \pm 73$ ms.

We have not precisely measured inference time for MDM and MLD. However, when generating motion and shape from noise, the largest time difference in the complete process, by far, between or

method and MDM/MLD is the step converting the network outputs to the mesh. As mentioned above, the SMPL-H model can generate the mesh from the SMPL parameters in about 31 ms. But MDM's and MLD's implementations of the SMPLify based post-hoc recovery take on the order of minutes.

## G LICENSES

1. EDM2 (Karras et al., 2024): Creative Commons BY-NC-SA 4.0
2. SMPL-H (Romero et al., 2017): See `https://mano.is.tue.mpg.de/license.html`
3. AMASS (Mahmood et al., 2019): See `https://amass.is.tue.mpg.de/license.html`
4. FID encoder (Guo et al., 2022): MIT license
5. HumanML3D (Guo et al., 2022): MIT license

## H LLM USAGE

The authors used ChatGPT 5 for the purposes of minor text polishing.

