# OpenReview forum: "Unconditional Human Motion and Shape Generation via Balanced Score-Based Diffusion"
_ICLR.cc/2026/Conference — Submitted to ICLR 2026_

### Official Review · Reviewer_2Ebj · 2025-10-30

**Soundness:** 3
**Presentation:** 2
**Contribution:** 2
**Rating:** 4
**Confidence:** 4

**Summary:**

The paper describes a new implementation for human motion generation, based on the insights from the EDM2 paper. The implementation has several merits, including less computational effort, less parameter tuning, and the incorporation of SMPL and shape parameters as opposed to recovering them from the joints. The paper also offers some theoretical analysis.
On the flip side, the proposed implementation does not present state-of-the-art results, has less capabilities, the removed computation is not shown to have merit, and evaluations are inconsistent and lacking.

**Strengths:**

- The paper incorporates a refreshing approach to human motion generation, different than that used by the community.
- SMPL-based prediction is interesting.
- Some insights, such as gradient balancing, seem quite relevant and important

**Weaknesses:**

- The paper shows partial capabilities - it did not incorporate any conditions (claiming this is "on purpose", but no benefit is shown), it does not demonstrate why or where the new approach and the capabilities it induces has merits.
- Results are below the state-of-the-art - the paper compares to two rather old models, which have both seen dozens if not hundred of improvements since their original publication. For example, in CAMDM and CLoSD, a diffusion model that uses only ±10 steps is used, with much lower FiD than the 1000 steps model compared against here.
- While shape is predicted, which is a refreshing plus, it doesn't seem to have any correlation to the motion itself, which makes shape prediction rather arbitrary.
- It is unclear what the contributions of the paper are. More than anything else, it shows that EDM2 has merit, but it is almost directly used in the context of motion generation. The "adaptations" from the original approach seem quite marginal. The paper should address better these adaptations and explain why they are not trivial. For example, dimension balancing, which seems to have a large effect on the results, is not adapted, while some more marginal parameters (such as tmin, which can be empirically found rather easily) are described in more details.

**Questions:**

- Perhaps the only analysis relevant in terms of design choice justification is Table 1, but it raises more questions than answers them. Why are the final numbers here different than those in Table 2? How would more conventional choices affect the model (e.g., adding redundancies to the representation / regularizing losses / text conditions)? Wouldn't it improve more? What is the baseline? It seems to be very poorly designed to show great improvement, but this does not justify that the design choices are indeed the right ones, just that they have some improvement.
- Why compare against old models?
- What about larger datasets that now available? Would this approach still have merit if trained on them?
- What is the usecase where this approach could shine the most?
- Could some of the insights (e.g. data normalization) be incorporated into traditional pipelines to improve SOTA?
- Auxiliary losses, which you claim to be unnecessary, help with things like jitter and ground penetration. How does this approach perform in these aspects? How much would the auxiliary losses help in this case?

---

> ### Author Response · Authors · 2025-11-18
> **Response to reviewer 2Ebj**
>
> Dear reviewer, we thank you for reading our paper and for the valuable feedback.
>
> To start, we would like to respectfully but firmly disagree with the assessment that our contributions “mainly show that EDM2 has merit” for human motion. Our baseline is a minimal adaptation of the EDM + EDM2 framework to human motion (i.e. 1D temporal data) with some updated hyperparameter choices. Table 1 shows that this baseline does not obtain good results. Karras et. al. (2024) introduced tools for balancing activations, neural network weights, and losses. Our methodological contributions, detailed in Section 4, consist of adapting and extending these tools to the domain of human motion. In particular, we use them for feature normalization, feature balancing and gradient balancing both over diffusion timesteps and between feature groups. Thus we have both extended the contexts in which these tools are applied, but also generalized, re-derived, and modified them. We also view the fact that these adaptions and extensions are conceptually simple, straightforward and effective as strengths of our method, not weaknesses. Below we will address the reviewers questions one by one.
>
> ### Questions
> __Ablations__
>
> We first would like to clarify why the numbers in Table 1 and Table 2 differ. As stated in Section 5.2 and detailed in Appendix A (Table 3), the model used during ablations is a smaller version of the final model. Concerning FID, the splits it is calculated on are also different.
>
> Quite possibly "more conventional choices" and hyperparameter tuning could improve the results. However, the primary goal of this work is not to minimize FID (or any other metric) at all costs. Our objective is to investigate whether results on par with state of the art can be achieved without these additional choices, while retaining PF-ODE compatibility and more. We believe our experiments demonstrate that this is indeed possible. They also show that feature and gradient balancing is important in much the same way that weight and activation balancing are crucial in EDM/EDM2 as shown by Karras et. al. (2024)
>
> Regarding the baseline, we disagree with the claim that it is "very poorly designed". As mentioned at the start of this response, The baseline is a straightforward instantiation of the EDM + EDM2 framework for human motion, using the same feature normalization scheme as used in the HumanML3D dataset. It is not tuned to be weak, and we believe it to be a reasonable baseline for assessing our proposed changes fairly.
>
> __Comparisons to prior work__
>
> We compare with MDM and MLD because (to the best of our knowledge) they are the only prior human motion diffusion works that evaluate unconditional motion generation. We would be very happy to made aware of other strong unconditional models we missed.
>
> __Larger datasets__
>
> We would love to train on larger high quality datasets. But we are not aware of which ones are publicly available. We anticipate our approach would benefit from more data. The method is not tied to a specific dataset and, if anything, its avoidance of empirically tuned loss weights should make it easier to apply to larger or newer datasets. We would therefore expect our approach to retain (and likely improve) its merits when trained on larger datasets.
>
> __Use case__
>
> A direct use case of our model is likelihood evaluation. In addition, it provides a strong, clean backbone on which to build conditional models. In our own follow-up work we will use this base model to condition on noisy and partially observed motions. In principle, any conditioning signal can be incorporated without changing the framework, as long as it is normalized to unit expected magnitude..
>
> __Insights and traditional pipelines__
>
> Some of the insights could in principle be incorporated into "traditional pipelines" but we believe that matters are complicated. For example, the over-parameterized representation in HumanML3D includes both 3D joint positions and joint angles, where the angles are only used for regularization (as stated by prior works). In such a setting, it is not obvious that applying our rotation normalization would be beneficial. Additionally, it is not obvious that balancing gradients equally is optimal in a setting where there are multiple representation of similar information, and only one of them is used after decoding.
>
> __Auxiliary losses__
>
> We claim that auxiliary losses should not be strictly necessary to match the human motion distribution with diffusion models, and our results support this. That said, several prior works have clearly demonstrated that such auxiliary terms are effective and it is not improbable that they could improve some metrics for our method as well. However, the cost is not insignificant as we loose PF-ODE compatibility and would need to do empirical tuning of their weights. We also think that adding auxiliary losses, while improving metrics that are measured, could add hidden biases that might go unnoticed.

---

### Official Review · Reviewer_D3V8 · 2025-10-31

**Soundness:** 3
**Presentation:** 3
**Contribution:** 3
**Rating:** 4
**Confidence:** 5

**Summary:**

This paper proposes a method for unconditional human motion and shape generation using a score-based diffusion model. The authors argue that common practices in this field—such as using over-parameterized feature representations (e.g., combining joint angles with 3D positions) and auxiliary losses (e.g., foot skating losses)—are unnecessary if the training process is properly balanced. They adapt the EDM2 framework to human motion, introducing strictly theoretically motivated normalizations for SMPL-based input features (specifically handling 6D rotations without breaking their structure) and deriving analytic weightings for the standard L2 loss to balance gradients across diffusion time-steps and different feature groups (joints, orientation, translation, shape).

**Strengths:**

1. The paper takes a refreshing "back to basics" approach. Instead of adding complexity to fix issues, it uses theoretical derivations to balance the standard training objectives.

2. Table 1 is very convincing. It clearly shows how each theoretical addition (input normalization, gradient analysis, per-group weighting, dimensionality addressing) cumulatively improves the baseline FID from 6.23 down to 2.40.

3. By staying strictly within the score-based diffusion framework without ad-hoc sampling guidance, the method retains compatibility with probability flow ODEs, allowing for deterministic sampling and likelihood evaluation.

**Weaknesses:**

1. The paper focuses solely on unconditional generation. While the authors argue this is a necessary first step, most practical applications require conditional generation (text-to-motion, action-to-motion). It remains to be seen if this "pure" balanced approach holds up when complex conditioning signals are introduced.

2. The evaluation is primarily on AMASS (HumanML3D subset). Broader evaluation might be needed to confirm the generalizability of these balancing terms across different motion datasets with different characteristics. There are more datasets for human motion as BABEL, Motion-X, Posescript, etc... an evaluation on those will give a better understanding of the method generalization abilities.

3. I appreciate the visual results attached to the paper, while their quality are not way better than current baselines.

**Questions:**

1. Have you performed any preliminary experiments on extending this balanced weighting scheme to conditional models (e.g., classifier-free guidance)? Does adding conditioning break the delicate balance achieved here?

2. MDM generates both 3D joint locations and joint rotations. If you used only their generated rotations with an SMPL model, it would also result in zero limb length variance. Why did you choose to compare against their 3D joint output (which has variance) instead of their rotation output for the "Limbo" metric?

3. Given that a person's shape ($\beta$ SMPL parameters) does not change during a motion, why does your per-frame representation in Eq. 9 include $\beta$? Appendix A.1 clarifies you just copy the same vector for each frame, but what is the motivation for this design over generating a single, time-independent shape vector for the entire sequence?

---

> ### Author Response · Authors · 2025-11-18
> **Response to reviewer D3V8**
>
> We would like to sincerely thank the reviewer for their time, feedback and positive assessment. Below we will answer the questions one by one followed by a brief comments on the suggested datasets and qualitative results.
>
> ## Questions
>
> __Conditional generation__
>
> We have not done any preliminary experiments using our framework in the conditional setting, however it is something that we plan to do in the near future and it is something that we have kept in mind while developing the current work. In principle nothing has to change in the conditional setting. Our balancing acts on the motion/shape features and the score-matching loss, which have the same form in the conditional case. Adding conditioning (text, actions, partial trajectories) does not require changing the loss or the weighting scheme. The requirement is that the conditioning features are normalized to unit expected magnitude. We have clarified this in the main text by writing a Future Work section.
>
> __MDM joint locations and rotations__
>
> It is correct that MDM generate both 3D joint locations and joint rotations, but there are two reasons we chose to use their 3D joint output for the Limb standard deviation metric. First in the HumanML3D representation the standard decoding procedure in prior work is to use the predicted 3D joint positions. The joint rotations are mainly described as being used for regularization, and there is little discussion or evaluation on their standalone quality. To keep the comparison faithful to how MDM is normally used and reported, we therefore relied on the 3D joint output. Secondly, the joint rotations in the HumanML3D representation are not SMPL joint rotations. They are derived from SMPL 3D joint locations and do not contain information about rotation around the limb axes. As a result, simply plugging these rotations into SMPL would not yield a meaningful, directly comparable SMPL pose with strictly fixed limb lengths.
>
> __SMPL shape parameters__
>
> There are mainly architectural and practical reasons for this design choice. Our goal was to keep the architecture as simple and close as possible to a standard per-frame sequence model, without including extra components such as separate shape encoder, global tokens, or explicit mechanism to broadcast shape information across frames. By copying the same shape vector to all frames, we can treat shape just like any other per-frame feature and reuse a straightforward architecture and training setup. Our results (both quantitative and qualitative) does seem to indicate that this design choice is fine. However, we still think this is something that would be interesting to investigate in future work.
>
> ### Suggested datasets
> We agree that BABEL could be interesting to evaluate on. However, since it is also built on AMASS, there is non-trivial overlap with HumanML3D. According to Li et. al. (2024), there is roughly 30% total overlap between the two datasets, and this is for the full HumanML3D set. Because we follow MLD and use only the AMASS subset of HumanML3D, the effective overlap is even larger. In the conditional case this would be more informative, since the labels differ more substantially, but we do not expect it to significantly change our understanding of our method's generalization ability in the unconditional setting. PoseScript contains individual poses but no motion, so it is not relevant to our work. We were not previously aware of Motion-X, which indeed appears to contain a significant amount of motion data beyond AMASS. Unfortunately, from a first look at the project page, many of these sequences appear to exhibit noticeable foot skating, an artifact we want to avoid.
>
> ### Qualitative results
> We agree that the qualitative results do not exhibit way better quality then the baselines. However, our motions (excluding rendering which is the same for all methods) take about 1.7s to generate while it takes on the order of minutes for MDM and MLD due to the slow post-hoc recovery of the mesh from 3D joint positions.
>
> _Chuqiao Li, Julian Chibane, Yannan He, Naama Pearl, Andreas Geiger and Gerad Pons-Moll. UniMotion: Unifying 3D Human Motion Synthesis and Understanding. arXiv preprint arXiv:2409.15904v2, 2024_

---

### Official Review · Reviewer_fc5w · 2025-11-02

**Soundness:** 2
**Presentation:** 2
**Contribution:** 2
**Rating:** 4
**Confidence:** 2

**Summary:**

This paper presents a principled score-based diffusion framework for unconditional human motion and shape generation. Unlike prior work relying on redundant input features (e.g., joint positions + velocities + foot contacts) and multiple auxiliary losses, this paper demonstrates that a balanced training strategy with theoretically grounded normalization and loss weighting is sufficient to achieve state-of-the-art performance.

**Strengths:**

1. This paper proposed a structure-preserving feature normalization across heterogeneous SMPL feature groups (pose, orientation, translation, shape);

2. The method removes the need for auxiliary losses and hand-tuned hyperparameters, simplifying the training pipeline. Implementation details are extensive and clear, making replication straightforward.

3. By avoiding auxiliary regularization, the model remains fully compatible with ODE-based sampling and likelihood computation—rare among motion diffusion works.

**Weaknesses:**

1. While generating motion and shape directly in the SMPL parameter space has the clear advantage of avoiding post-hoc shape reconstruction, the SMPL parameters themselves are typically obtained by fitting to skeletal joint data. This fitting process can introduce non-negligible errors (e.g., local rotation ambiguity, imperfect alignment), which may propagate into the generative model and limit the ultimate fidelity of the generated results.

2. The paper introduces a separately trained auxiliary network to learn dynamic weights for different feature groups. Although the authors claim that this network is lightweight, the paper does not provide quantitative comparisons (e.g., training time per epoch, convergence speed) to verify that the added module does not increase the overall training cost.

3. The proposed model focuses solely on unconditional motion and shape generation. While unconditional modeling can provide a reliable prior over motion distributions, most real-world applications (e.g., text-to-motion, action-conditioned motion synthesis) require conditional generation. Extending the framework to such settings would significantly enhance its practical impact.

**Questions:**

Please refer to the weaknesses above.

---

> ### Author Response · Authors · 2025-11-18
> **Response to reviewer fc5w**
>
> We would like to sincerely thank the reviewer for their thoughtful and constructive feedback, as well as for the positive assessments of our contributions. Below we address the concerns point by point.
>
> __SMPL parameter space__
>
> This is a good point, and we agree the SMPL fitting is not error-free. There are a few aspect we would like to clarify. First, it is fairly common that SMPL parameters are estimated from both joint and image information (monocular or multi-view), which helps resolve some of the local ambiguities you mention. Second, we do not see this situation as fundamentally different from the 3D joint case, those joints are also typically estimated through noisy processes (marker-based or markerless optical tracking, VICON, etc.). Concretely for HumanML3D, there is in fact no difference between our method and the baselines in terms of upstream fitting noise, because the 3D joints used in the HumanML3D representation are themselves derived from SMPL parameters. More broadly, most generative works implicitly assume access to "clean enough" samples from the target distribution, except works where learning from noisy observations is the explicit focus. Given that truly clean human data is difficult to come by in any parameterization, we agree that developing methods that account for noise is an interesting direction for future research. Additionally, for applications that require the full mesh (shape and rotations around limb axes), any approach that predicts only 3D joints will still need a post-hoc mesh reconstruction step, which reintroduces exactly the kind of ambiguities and fitting errors discussed above. Our choice to work directly in SMPL space moves this fitting step "offline". Once the dataset has been prepared and the model trained, inference becomes much faster and avoids repeated per-sample reconstruction at test time. Finally, if this is still a concern, we do report on par with state of the art results using a 3D joint parameterization (Ours Root rel. in Table 2) as well.
>
> __Auxiliary network.__
>
> The auxiliary network (which we have four versions of, one for each feature group) consists of Fourier features, a single linear layer, and a scalar gain. Its parameter count and computation are negligible compared to the denoising network. This is why we initially did not report separate timing numbers. For completeness, we have now measured the average run time of a forward and backward pass, which takes about 1 ms per auxiliary network and batch with a batch size of 64. We have added a short remark regarding this in the end of Section 4.5.
>
> __Conditional generation__
>
> We agree that unconditional models have less direct, standalone practical impact than conditional ones. However, we do not believe this diminishes the value of the insights from a research perspective. This project originally started from a conditional use case. We were interested in motion infilling with large temporal gaps. In exploring this, we were not satisfied with the results and concluded that we needed to look more closely at unconditional priors first. Conditional use cases have therefore been a central motivation throughout this work, not an afterthought. Importantly, nothing in our approach is specific to the unconditional setting. Our balancing acts on the motion/shape features and the score-matching loss, which have the same form in the conditional case. Adding conditioning (text, actions, partial trajectories) does not require changing the loss or the weighting scheme. The requirement is that the conditioning features are normalized to unit expected magnitude. We have clarified this in the main text by writing a Future Work section.

---

### Official Review · Reviewer_Aswg · 2025-11-04

**Soundness:** 3
**Presentation:** 3
**Contribution:** 3
**Rating:** 8
**Confidence:** 3

**Summary:**

This paper proposes a balanced score-based diffusion framework for unconditional human motion and shape generation, which eliminates heuristic losses and feature redundancy by introducing theoretically derived normalization and loss balancing across heterogeneous feature groups. The key contributions are:
1. Unified diffusion formulation that directly models both human motion and shape using SMPL parameters.
2. Analytical feature balancing mechanism that replaces empirical weighting with theoretically grounded normalization.
3. Training framework compatible with PF-ODE sampling, achieving strong performance with only 31 neural function evaluations (NFEs).
4. Comprehensive evaluation and ablation showing improved fidelity, diversity, and efficiency over state-of-the-art diffusion-based motion models.

**Strengths:**

Originality is strong. The paper presents a conceptually fresh rethinking of how diffusion models are trained for motion generation. Instead of stacking auxiliary losses or handcrafted feature weights, it introduces a principled analytical balance across motion and shape components. This approach reduces reliance on empirical tuning while maintaining state-of-the-art performance, which is a notable departure from prior diffusion-based motion generation pipelines.

Quality is good. The method is both theoretically motivated and empirically validated. Ablation studies systematically isolate the effect of feature normalization and balanced weighting, and the model demonstrates strong results on multiple benchmarks (FID, diversity, and smoothness). However, the paper lacks deeper theoretical guarantees (e.g., convergence proof) and broader task evaluations beyond unconditional generation, leaving some generalization aspects unexplored.

Clarity is strong. The paper is clear, well-organized, and mathematically precise. Each design choice is motivated and supported by quantitative evidence. Equations and visual diagrams are intuitive, and the narrative flows smoothly from motivation to results. The supplementary materials provide necessary implementation details, enhancing reproducibility.

Significance is strong. The work contributes to a key ongoing trend: efficient and interpretable diffusion modeling. Its training simplifications (analytical balancing, PF-ODE compatibility) make diffusion-based motion synthesis more practical and theoretically consistent. These ideas can influence future research in both motion generation and general generative modeling.

**Weaknesses:**

Lack of formal analysis

The proposed balancing scheme is inspired by prior theoretical insights but lacks formal convergence or stability proofs.

Suggestion: Include a mathematical analysis (or at least empirical sensitivity tests) to show that the balance remains stable across datasets or motion scales.

Narrow experimental scope

Only unconditional motion is evaluated.

Suggestion: Extend to conditional setups (text-to-motion, action labels) or discuss how the framework could generalize to such tasks.

Computational evaluation missing

Efficiency claims focus on NFEs but omit runtime, training cost, and GPU memory comparisons.

Suggestion: Add computational benchmarks to support claims of superior efficiency.

Lack of variance reporting

Metrics like FID and diversity are reported as single values without standard deviation.

Suggestion: Provide averaged results over multiple seeds with confidence intervals.

Data assumptions

The method assumes access to full SMPL parameters.

Suggestion: Discuss adaptability to datasets with partial 3D keypoints or noisy inputs.

**Questions:**

1. How sensitive is the model to the specific analytical weight formulation? Could learned or adaptive weight schedules perform better?

2. Does the proposed balancing framework generalize to conditional diffusion models (e.g., text or action-conditioned)?

3. How does the model handle longer or variable-length motion sequences?

4. How robust is the model to missing or noisy SMPL parameters at inference time?

5. Would the removal of auxiliary losses reduce diversity in rare or extreme motion types?

---

> ### Author Response · Authors · 2025-11-18
> **Response to reviewer Aswg**
>
> Thank you for your detailed and positive assessment of our work. We appreciate the recognition of its  originality,
> clarity and significance. We now address the specific issues raised.
> ### Questions
> __Learned or adaptive weight schedules__
>
> The weightings of the different elements (diffusion time step and feature group) of the L2 loss used to train the denoising network, see the denominator within the expectation of equation (20), are adaptive and learned, but have analytical expressions. Each weight is based on the scalar output of the lightweight uncertainty weighting network (one per feature group) which takes the diffusion time step as input. The parameters of each uncertainty network are updated during training, using mini-batch gradient descent, to minimize the loss defined in equation (15). The effectiveness of our approach therefore relies on the convergence of these uncertainty networks to reasonable values. We have not yet tried on another dataset. However, a related formulation has been tested on ImageNet by Karras et al. (2024). This part of our contribution changes how the learned value is used, i.e. to balance gradients instead of loss and to balance over feature groups as well over diffusion time steps, while the learning mechanism is minimally changed.
>
> __Generalization to conditional generation__
>
> Yes, our approach generalizes to conditional diffusion models. Our balancing acts on the motion/shape features and the score-matching loss, which have the same form in the conditional case. Adding conditioning (text, actions, partial trajectories) does not require changing the loss or the weighting scheme. The requirement is that the conditioning features are normalized to unit expected magnitude. We have clarified this in the main text by writing a Future Work section.
>
> __Variable length__
>
> The model is trained on variable length sequences, at test time it should handle shorter sequences within the trained range. We have not yet systematically evaluated performance w.r.t. the sequence's length. But from qualitative experience we see generally that shorter sequences are easier to model and longer ones are harder. There is nothing in the architecture that prevents longer generation, the backbone is mainly convolutional over time so in principle it can be run on longer inputs. However, for lengths significantly beyond the training regime, we would expect quality to degrade. A more thorough analysis of very long-horizon generation is an interesting direction for future work.
>
> __Missing or noisy SMPL parameters__
>
> For unconditional generation, there is no notion of missing or noisy SMPL parameters at inference time, since the model generates the full sequence SMPL parameters directly from Gaussian noise. In future work, we plan to build on this current approach to generate motions conditioned on noisy and partial observations.
>
> __Auxiliary losses__
>
> We agree auxiliary losses might help penalize generating physically implausible motions, but they could also discourage generating rare but valid ones and reduce diversity. The results in Table 2 could indicate this, as our method demonstrates higher diversity.
>
> ### Suggestions
> __Mathematical analysis / empirical sensitivity test__
>
> We are not entirely sure which specific form of additional mathematical analysis or empirical sensitivity test the reviewer has in mind and ask for clarification. We are happy to add things to further show the generalization of our method.
>
> __Efficiency claims__
>
> Our main claim is that we avoid the expensive post-hoc mesh recovery from 3D joints, which takes on the order of minutes per sequence, whereas our model produces the parameters to generate the mesh quickly using the SMPL-H model. We have added a paragraph at the end of Appendix F to clarify this. We have kept the NFE column in Table 2 for context, but removed the bold font. Regarding training efficiency we have not done any careful benchmarking. However, neither our method nor MDM/MLD require massive amounts of computation or memory, all capable of being trained on consumer GPU cards. Our final model trained on a NVIDIA RTX 3090 took ~28 hours and required ~7.2 GB VRAM.
>
> __Variance reporting__
>
> We follow the evaluation protocol of Karras et. al. (2024). Each evaluation is run three times. One evaluation corresponds to generating N samples and then using these samples to compute the quantitative metrics. N is chosen to a number of samples to ensure low variation in all the metrics between all the 3 evaluation runs (N = 5000 in our case). This procedure results in low standard deviations, for example the final models FID and foot skating standard deviations both are 0.05. However, we noticed a mistake in Appendix A.6 where we write that the variation is less than 2%, which is not strictly true for the final model evaluated on the test set which has a variation of about 2.6% on the FID metric. We have changed Appendix A.6 to reflect this.

---

### Meta-Review · Area_Chair_rYoZ · 2026-01-05

**Summary:**

[Strengths]

- The paper provides a principled, theoretically grounded reformulation of score‑based diffusion for human motion, including expected‑magnitude normalization and gradient balancing across both diffusion timesteps and SMPL feature groups. (Reviewer fc5w, D3V8)
- The method preserves PF‑ODE compatibility, enabling deterministic sampling and likelihood evaluation—an uncommon but valuable property in human motion diffusion models. (Reviewer fc5w, D3V8)
- The implementation is clean and reproducible, with clear ablations (Table 1) and detailed training and architectural descriptions. (Reviewer fc5w, D3V8)

[Weaknesses]

- The work is limited to unconditional generation, without any empirical demonstration that the proposed balancing scheme extends to practical conditional tasks (e.g., text‑to‑motion). This significantly limits impact. (Reviewer fc5w, D3V8, 2Ebj)
- Evaluation scope is narrow: experiments rely solely on HumanML3D/AMASS, and comparisons do not include more modern strong baselines, leaving generalization and competitiveness uncertain. (Reviewer D3V8, 2Ebj)
- The improvements appear incremental; the final performance does not surpass state‑of‑the‑art, the design motivation for certain choices (e.g., per‑frame β) is not fully convincing, and the empirical evidence for the method’s practical advantages is limited. (Reviewer 2Ebj)

**Reviewer Concerns:**

[Addressed by rebuttal]

- The computational overhead of the auxiliary per‑group weighting networks was clarified (≈1 ms per network per batch), alleviating concerns about added training cost. (Reviewer fc5w)
- The choice to compare MDM using its 3D joint output rather than rotations was justified, as HumanML3D’s rotation representation does not correspond to SMPL axis‑angle rotations and cannot be plugged into SMPL consistently. (Reviewer D3V8)
- The design choice of copying the shape vector beta across all frames was explained as an architectural simplification compatible with the 1D U‑Net, with no observed negative effects. (Reviewer D3V8)

[Still outstanding]

- The absence of conditional experiments remains the core limitation; despite theoretical arguments, no empirical validation was provided. (Reviewer fc5w, D3V8, 2Ebj)
- Generalization and competitiveness were not demonstrated: the paper does not test on other datasets (e.g., BABEL, Motion‑X) nor compare against more recent baselines (e.g., CAMDM, CLoSD). (Reviewer D3V8, 2Ebj)
- Practical impact concerns persist: insufficient evidence that the proposed framework improves shape‑motion correlation, contact quality, or other application‑critical behaviors. (Reviewer 2Ebj)

**Reviewer Scores:**

- Reviewer Aswg: 8
- Reviewer fc5w: 4
- Reviewer D3V8: 4
- Reviewer 2Ebj: 4

---

### Decision · Program_Chairs · 2026-01-26

Reject